# Mutual Effect of Components of Protective Films Applied on Copper and Brass from Octadecylamine and 1,2,3-Benzotriazole Vapors

**DOI:** 10.3390/ma15041541

**Published:** 2022-02-18

**Authors:** Olga A. Goncharova, Andrey Yu. Luchkin, Nina P. Andreeva, Vadim E. Kasatkin, Sergey. S. Vesely, Nikolay N. Andreev, Yurii I. Kuznetsov

**Affiliations:** Alexander Naumovich Frumkin Institute of Physical Chemistry and Electrochemistry, Russian Academy of Sciences, Leninskii Pr. 31, 119071 Moscow, Russia; skay54@yandex.ru (A.Y.L.); andrnin@mail.ru (N.P.A.); vadim_kasatkin@mail.ru (V.E.K.); sergei57@mail.ru (S.S.V.); n.andreev@mail.ru (N.N.A.); yukuzn@gmail.com (Y.I.K.)

**Keywords:** copper, brass, atmospheric corrosion, corrosion inhibitors, chamber corrosion inhibitors, mixed corrosion inhibitors, mutual effect of components

## Abstract

It has been shown by a set of corrosion, electrochemical and physical methods that a chamber corrosion inhibitor that consists of a mixture of octadecylamine (ODA) and benzotriazole (BTA) efficiently protects copper and brass from atmospheric corrosion and can be used for the temporary protection of metal items. The optimum temperatures of treatment with the ODA + BTA mixed inhibitor is 120 °C for brass and 100 °C for copper. One-hour treatment in ODA + BTA vapors at these temperatures results in the formation of nanosized adsorption films on the surface of these metals. These films stabilize the passive state and provide efficient temporary protection of metal items. The ODA + BTA inhibitor is superior to its components in terms of protective aftereffect. Our analysis of the mutual effect of BTA and ODA indicated that they show an antagonism of protective action on copper, but there is also a synergistic enhancement in the case of brass. Electrochemical impedance spectroscopy studies demonstrate that the inhibitors in question mainly act by using a blocking mechanism on copper and brass. Chamber treatment of the metals studied in vapors of the ODA + BTA mixture resulted in a noticeable hydrophobization of the copper surface and an insignificant effect on the brass surface. Chamber treatment of copper samples with artificially created polymodal roughness made it possible to obtain a superhydrophobic surface.

## 1. Introduction

The vapor-phase protection of metals with inhibitors [1,2,3,4,5,6,7,8,9,10,11] is a reliable way for the preservation of metal items. Volatile corrosion inhibitors (VCIs) are widely used in practical applications. Agents of this type were developed in the 1940s, and studies aimed at their improvement continue to date [1,2,3,4,5,6]. Chamber inhibitors (CINs) are not so well-known. The theory of action of CINs was founded by the authors of this article and has been developed in the past five years [7,8,9,10,11]. The pros and cons of VCIs and CINs are described in detail in Reference [12]. Commercial CINs are now only beginning to appear on the market as means for temporary protection. Moreover, CINs that protect a few metals at once have the best prospects of industrial application. A mixture of octadecylamine (ODA) and 1,2,3-benzotriazole (BTA) belongs to formulations of this kind.

The protective properties of this CIN on steel was studied previously [12]. In the same work, the following is noteworthy:-It was shown that a CIN comprising an ODA + BTA mixture efficiently protects steel from atmospheric corrosion and can be used for the temporary protection of metal items;-The optimal temperature of chamber treatment (T_CT_) and its duration (τ_CT_) required to form nanoscale adsorption films on the steel surface in order to stabilize the metal passive state were determined;-It was found that the ODA + BTA mixture is superior to ODA or BTA alone in the protective aftereffect (PAE), but estimation of the mutual effect of the mixture components indicates their antagonism;-The protection of steel with ODA, BTA and their mixtures was found to occur by a mixed (blocking-activation) mechanism;-It was shown that chamber treatment (CT) of steel with an ODA + BTA mixture resulted in surface hydrophobization or superhydrophobization if a polymodal surface was created on steel before the CT.

The materials presented in this article continue the previous publication [12], are closely related to it, rely on the theoretical concepts put forward in it and deal with a study of the protective properties of the ODA + BTA combination and its components toward copper and brass.

## 2. Materials and Methods

Samples and electrodes of M1 copper and L62 brass were used in the experiments. Their composition is defined in the corresponding standards [13,14]. The experimental techniques used were identical to those reported previously [12]. In addition, the protective properties of the CIN were studied in a salt fog chamber (SFC). These experiments were carried out at room temperature in an WEISS SC/KWT 450 chamber (WEISS Technik, Germany). The number of cycles until the appearance of corrosion damage on the samples was recorded in the tests. Each cycle included a 15-min spraying of 3% sodium chloride solution, followed by 45-min exposure of the specimens in the resulting salt fog. The samples were examined after each cycle.

## 3. Results

### 3.1. Copper

#### 3.1.1. Accelerated Tests

The optimal conditions for the chamber treatment of copper (T_CT_ and τ_CT_) were determined in two series of accelerated tests, namely in a KTV-0.1-002 heat-and-moisture chamber (HMC) (OJSC “KlimatSpetsMash”, Volgograd, Russian Federation), in which 100% relative air humidity was supplemented by daily moisture condensation on the samples, and in a salt fog chamber. Thermal treatment (TT) of copper samples without inhibitors (blank test) did not affect the protection time (τ_prot_) under recurrent condensation conditions (Table 1). After 4 h of exposure in a corrosive environment, dark spots appeared on the copper. It is essential to note that one-hour exposure of copper samples at a temperature of 120 °C or higher changed the color of samples, due to metal oxidation.

The adsorption films of ODA and BTA significantly inhibited the beginning of corrosion, even at T_CT_ = 80 °C. The first signs of corrosion damage on copper appeared after 288 h (ODA) or after 456 h (BTA). A longer protective period (3576 h) under these conditions is provided by films obtained in vapors of the ODA + BTA mixture.

As one can see from Table 1, a T_CT_ increase to 100 °C improved the protective effect of the adsorption films. The value of τ_prot_ increased twofold after treatment in BTA vapors, or threefold after treatment in ODA vapors. In the case of the ODA + BTA mixture, the PAE increased 1.3-fold. The first traces of corrosion damage appeared on copper samples in 960 h after the start of the experiment with BTA, after 1080 h with ODA or after 4800 h with ODA + BTA.

If T_CT_ was increased to 120 °C or higher, undesirable darkening of the copper surface occurred, both after treatment with inhibitors and without them. Therefore, the subsequent study was limited to CT in the temperature range of 80–100 °C.

Thus, T_CT_ = 100 °C is the optimal temperature for the protection of copper with the CIN mixture. As in the case of steel 3 [12], the CIN mixture provided more efficient copper protection compared to its components. However, the calculation of the coefficient of mutual effect of the mixture components α introduced in the previous publication [12] indicates an antagonism of BTA and ODA at 100 °C:α = τ_prot, ODA + BTA_^meas^/τ_prot, ODA + BTA_^calc^ = 0.02 (1)
where the “meas” and “calc” indices refer to the protection times determined experimentally and are calculated while assuming that no mutual effects of components exist:τ_prot, ODA + BTA_^calc^ = τ_prot, ODA_^meas^ τ_prot, BTA_ ^meas^/τ_prot, 0_
(2)

The protection of copper by the CIN were studied by using T_CT_ = 100 °C and τ_CT_ = 60 min. The first corrosion damage on copper samples not treated in CIN vapors appeared in the SFC after 2 h of testing (Table 2), and local corrosion was observed.

Even CT for 20 min at 100 °C resulted in significant (7–7.5 fold) inhibition of copper depassivation. All the CIN studied showed approximately the same efficiency. The corrosion pattern was the same after chamber treatment with any of them.

An increase in CT duration from 20 to 60 min in BTA or ODA + BTA vapors almost did not affect the PAE of the adsorption films. Only treatment with ODA vapor resulted in a monotonic PAE increase with an increase in the CT time. Table 2 shows that, with a one-hour treatment of copper in ODA vapors, the PAE reached 24 cycles; that is, it increased by 70%.

#### 3.1.2. Ellipsometry

It should be noted that CT with ODA vapors demonstrated a better protective effect in SFC tests of copper compared to BTA or ODA + BTA vapors. Based on the corrosion tests described above, this CIN by itself can be used for efficient protection of copper. However, ODA used alone is not so versatile, since it is inferior to the ODA + BTA mixture in the protection of steel [12] and brass, as shown below. As concerns the mixed CIN, the data given in Table 2 indicate that one-hour CT is certainly sufficient for the formation of protective layers with the maximum PAE for copper.

The ellipsometric estimates of the thickness of films formed on copper upon treatment in CIN vapors are given in Table 3.

In the original state of copper before heat treatment, the air-formed oxide was 1 nm thick. Heat treatment in the absence of a CIN the increased thickness of the oxide film to 4–5 nm. Treatment of copper in CIN vapors decreased the thickness of the oxide film two- to three-fold and, at the same time, resulted in the formation of rather thick surface adsorption films of the CIN. As one can see from Table 3, the thickness of these films increases in the series BTA < ODA < ODA + BTA. These CIN films noticeably inhibit the growth of the air-formed copper oxide during the chamber heat treatment and directly affect the increase in the protective aftereffect of inhibitor-treated copper.

#### 3.1.3. Voltametric Experiments

Figure 1 presents the anodic polarization curves on copper samples with various methods of surface modification with the CIN or without treatment. The characteristic potentials obtained in these experiments are summarized in Table 4. The shape of all the polarization curves is characteristic of the passive metal in a certain potential range before reaching *E*_pit_ and/or the passive film breakdown potential *E*_br_ under considerable anodic polarization. The corrosion potential, *E*_cor_, of copper samples shifts cathodically after treatment with inhibitor vapors. The individual components (BTA and ODA) give nearly the same shift in *E*_cor_ (about −70 mV), while the mixed inhibitor shifts the open-circuit potential by −560 mV.

The anodic polarization of samples that were not treated with a CIN resulted in a gradual current increase. Minor current oscillations were observed above *E*_pit_ = 0.270 V, due to the beginning of surface oxide degradation, and film breakdown occurred at *E* = 0.510 V. Local dissolution pits were visible on the surface after removal of electrodes from the cell in the region of current oscillations.

Copper samples that underwent CT with inhibitors showed nearly no noticeable current oscillations during the anodic polarization until the film breakdown occurs. Compared to the “blank sample”, treatment with the CIN increased the breakdown potential considerably (by 500–700 mV). The greatest effect was obtained for the sample treated in ODA vapors. This effect of a sharp increase in *E*_br_ unambiguously indicates that the protective properties of the surface oxide on copper increase significantly after modification in CIN vapors.

The conclusions obtained by analysis of polarization curves match the results of testing the corrosion resistance of copper in a salt fog, where the highest protection efficiency was also achieved after treatment in ODA vapors. Though the *E*_pr_ of copper was lower after treatment with ODA + BTA vapors than in the case of ODA, the anti-pitting basis of the mixed inhibitor was larger.

#### 3.1.4. Electrochemical Impedance Spectroscopy (EIS)

Figure 2 shows the Nyquist plots of copper samples after CT without and in the presence of CIN. Impedance measurements were performed in potentiostatic mode at open circuit potentials for each sample. All the hodographs are slightly deformed semicircles with varying curvature depending on frequency. The presence of at least two time constants in the frequency spectra allows this system to be simulated by using an equivalent circuit similar to the one we used before [12]. The mentioned article provides a detailed discussion of the interpretation of its elements, which is traditional for corrosion studies of metals with film layers.

The values of the equivalent circuit elements calculated by simulation of the experimental results are presented in Table 5. The deviation S between the experimental impedance points and those calculated by the model did not exceed 4%. Turning to the analysis of the results obtained, it should be noted that the R_s_ parameter refers to the properties of the electrolyte and the cell, and its value is the same in all the cases.

In this model, the R_sl_ parameter characterizes the resistance that appears in the surface layer to the transfer of ions from the near-electrode solution layer to the metal surface (and back). This layer is a thermal air-formed oxide in the “blank experiment” and the oxide modified with CIN adsorption films. In these experiments, we failed to separate the two sublayers (in contrast to in ellipsometry). However, one can see that, in comparison to a sample that did not undergo treatment with an inhibitor, the use of a CIN leads to a sharp increase in R_sl_.

If we estimate the coefficient of corrosion inhibition because of surface blocking (γ_sl_) as the ratio of R_sl_ resistances of a sample after CT and after thermal treatment without a CIN, we get the following:γ_sl_ = R_sl_^CT^/R_sl_^TT^, (3)
where one can see that the protective properties of the surface layer increase almost 6-fold in the following series: BTA < ODA < ODA + BTA. Inhibited samples show a nearly linear relationship between the thickness of the adsorption film and the resistance of the surface layer.

The R_ct_ parameter (The R_ct_ values, similar to the other parameters in the table, are given with respect to the geometric surface area of the sample) refers to the charge transfer resistance in the Faraday corrosion process on the active areas of the metal surface (in this case, copper). This factor also increases sharply if a CIN is used, but in contrast to R_sl_, no correlation with the film thickness derived from the ellipsometric data is observed. The inhibition coefficient of the charge transfer Faraday process is shown in Table 6:γ_ct_ = R_ct_^CT^/R_ct_^TT^
(4)
where the increase is as follows: ODA < ODA + BTA < BTA.

The capacitive characteristics of the system can be estimated from the modulus and phase factor of the constant phase elements in our model. Here, CPE_sl_ is responsible for the geometric capacitance of the sample whose value is determined by a combination of the thickness and dielectric properties of the surface layer, the roughness and/or homogeneity factor, and the diffusivity of the liquid part of the capacitor plate. For all the samples, this element can be considered a fairly perfect capacitor whose specific capacitance consistently decreases in the series “Without CIN” > BTA > ODA > ODA + BTA, i.e., antibately to the surface film thickness. This confirms the strong surface blocking effect of the CIN found in the analysis of R_sl_ values.

The CPE_dl_ element characterizes the properties of the double-layer capacitance of the electrochemical reaction itself on the active surface areas. A decrease in the CPE_dl_ modulus is apparent. Its value decreased more than tenfold for samples treated with a CIN compared to the samples that underwent heat treatment without inhibitors. It may indicate that inhibition of the reaction and blocking of the active surface occur. This effect is most noticeable in the case of BTA, where the phase factor n = 0.79 indicates a noticeable heterogeneity of the active sites.

Thus, the results of experimental data simulation by the equivalent circuit for copper samples treated with CIN allow one to assume that the CIN studied predominantly act on copper by using a blocking mechanism. As one can see from Table 5, the overall protective efficiency, Z, determined from electrochemical impedance data increases as follows: BTA < ODA < ODA + BTA, and it is about 90% for all the inhibitors.

#### 3.1.5. Contact Angle Measurement

The results of the contact-angle measurement under optimal conditions of copper CT by the inhibitors studied are given in Table 7.

Similar to the case of steel, copper treatment by the CIN resulted in the hydrophobization of its surface. After CT in BTA or ODA vapors, the values of θ acquired were 107 and 104°, respectively. The maximum hydrophobization of copper (θ = 112°) was obtained with the ODA + BTA inhibitor. The same inhibitor gave a superhydrophobic surface (θ = 160°) on copper with preformed polymodal roughness.

The above data showed that the mixed CIN was highly efficient and gave us a reason for testing it under natural conditions. Taking into account the experience obtained in similar tests on steel [12], we placed copper samples in a cardboard box to protect the metal from the deposition of corrosive compounds from the atmosphere.

The first indications of corrosion on copper samples that were not subjected to CT appeared after 20 days in the urban atmosphere. The CT of the metal with the ODA + BTA mixture gave total protection of the samples from atmospheric corrosion for a period longer than 25 months, which is quite sufficient for temporary protection. At the time this article was written, the copper samples showed no signs of corrosion, and their exposure under natural conditions continued. Pictures of samples tested for 25 months under the above conditions are presented in Figure 3.

### 3.2. Brass

#### 3.2.1. Accelerated Corrosion Tests

Brass underwent local corrosion under recurrent moisture condensation. Within 2 h of exposure of the alloy under the test conditions, black dots appeared on the samples that did not undergo thermal treatment. The corrosion initiation time did not depend on the alloy treatment temperature.

The protective properties of the adsorption films formed on brass at 80 °C in BTA or ODA vapors did not exceed the background values, but the BTA + ODA mixture provided brass protection for 24 h under these conditions. At *t*_CT_ = 100 or 120 °C, ODA did not protect brass, while BTA protected it very weakly by slowing down the formation of the first pits 2.5-fold. However, the ODA + BTA mixture demonstrated very efficient protection under these conditions. The films formed in its vapors at 100 °C completely prevented brass corrosion for 1416 h, or for 2232 h after treatment at 120 °C.

CT of brass in the vapors of the inhibitors studied at 140 °C, similar to TT of samples without a CIN, resulted in brass tarnishing. Therefore, these samples were not tested any further.

Thus, T_CT_ = 120 °C is the optimal temperature for the CT of brass in ODA + BTA vapors. Table 8 demonstrates that a significant synergistic effect is observed between the components of the ODA + BTA mixture. The coefficient of mutual effect of the mixture components exceeded 446. The mechanism of this synergy is not entirely clear and requires additional studies that are beyond the scope of this work.

Similar to copper, the optimal τ_CT_ was determined by testing brass in a salt fog chamber. The first corrosion damage on samples not treated in CIN vapors appeared after one test cycle (Table 9). The corrosion was of a pitting nature.

In the tests with BTA or ODA at 120 °C, increasing the CT time from 15 to 30 or 60 min did not slow down the metal depassivation. The CT of brass samples with the mixed CIN for the same period of time noticeably slowed down corrosion initiation to 3, 5, or 15 cycles, respectively. However, the character of brass corrosion did not change.

Heat treatment of brass at 120 °C for more than 60 min caused tarnishing of the alloy, regardless of whether the CIN studied was present. Thus, the duration of brass CT with a mixed CIN at the optimum temperature should not exceed 60 min.

#### 3.2.2. Ellipsometry

The results of ellipsometric estimation of the thickness of films formed on brass upon treatment in various modes are given in Table 10.

After heat treatment of brass without a CIN, an oxide film up to 3–3.5 nm thick was found on the surface. It is evident from Table 10 that the thickness of the oxide decreases somewhat after the CT, but the CIN films formed significantly exceed in thickness both the oxide film and the adsorption films of inhibitors on steel [12] or copper.

The thinnest adsorption film (~20 nm) was detected after treatment with BTA. The layers of ODA and the ODA + BTA mixture were almost two times thicker (35–40 nm). They were by an order of magnitude thicker than those formed on copper by the same CIN. Obviously, in this case, the physicochemical nature of the alloy has a decisive effect. However, the thickness of adsorption films on brass does not prevent the use of the mixed CIN for the preservation of precision brass items, where the tolerance of the geometric dimensions is very small, since the films are hundredths of a micron thick.

#### 3.2.3. Voltametric Experiments

The anodic polarization curves of brass electrodes (Figure 4) had an extended passive region, followed by a region of insignificant current oscillations and gradual current increase in the potential range of 0.9–1.3 V. There was no sharp increase in current density on the polarization curves accompanied by an inflexion on the curve (due to breakdown of the protective film), in contrast to the experiments with copper. Black pits could be observed on the electrodes after the anodic polarization curves were recorded.

The characteristic corrosion potentials, *E*_cor_, and potentials of the beginning of intense dissolution with formation of pits, *E*_pit_, are summarized in Table 11.

One can see from the table that the *E*_cor_ values of all brass samples are around 0.1 V, except that after treatment in ODA + BTA vapors, where *E*_corr_ increases by 40–50 mV. This behavior sharply differs from similar tests with copper, where the mixed CIN strongly shifted *E*_cor_ in the cathodic direction.

The potentials above, for which a fast increase in current densities was observed listed in the table as *E*_pit_, consistently shift anodically after treatment with CIN vapors. Most likely, this potential is associated with the beginning of surface film destruction, as well as intensification of the anodic dissolution of the alloy base. Moreover, the *E*_pit_ values are similar for the individual CIN but markedly more positive (by 0.4 V) for their mixture.

Thus, all the CIN studied meet the criteria of inhibitory protection if the chamber treatment of brass is performed under the optimal conditions. The mixed inhibitor is much superior to its components in terms of protective effect (higher overvoltage of the anodic reaction).

#### 3.2.4. Electrochemical Impedance Spectroscopy

Figure 5 shows the hodographs of brass specimens subjected to chamber treatment under the optimal conditions. The characteristic shape of the hodographs, i.e., deformed semicircular arcs, allows this system to be simulated by using the same equivalent circuit as discussed above. The error S between the experimental and calculated hodographs obtained in experiments on brass followed by computer simulation with optimization of the circuit elements did not exceed 5%. The calculated values of the circuit elements are presented in Table 12.

Since a scheme of the same type was used, let us apply a similar approach and analyze the protective effects of the CINs by the blocking and activation mechanisms summarized in Table 13.

It is obvious from the data presented above that BTA gives the smallest effect among the inhibitors studied. This is manifested in the degrees of protection both by the blocking and activation mechanism. Nevertheless, the comparison of these parameters to those of copper shows that this inhibitor protects brass much more efficiently and these parameters are quite comparable with those of St3 [12].

Much higher γ_sl_ and γ_ct_ values were achieved for ODA, but the ODA + BTA mixture gave the best results. This CIN has a strong blocking effect (γ_sl_ = 1609) and, at the same time, provides a high degree of inhibition of the Faraday corrosion process (γ_ct_ = 866). This behavior well correlates with the fact that the oxide-adsorption layer formed by ODA + BTA on the surface of brass is considerably thicker than on the other metals studied.

The geometrical capacitance, CPE_sl_, of brass specimens after treatment with CIN is an order of magnitude smaller than in experiments on steel or copper, probably also due to a thicker coating. The Faraday capacitance CPE_dl_ for CIN on brass is by an order of magnitude lower than on copper and by two orders of magnitude smaller than on steel. This also confirms a high degree of blocking of the active centers of the anodic reaction.

Following the logic used in the interpretation of EIS results on steel and copper, it can be stated that BTA predominantly acts on brass by using the blocking mechanism. The protective efficiency of CIN increases in the series BTA < ODA < ODA + BTA, thus confirming the results of corrosion and potentiodynamic experiments.

#### 3.2.5. Wetting Angle Measurement

The wetting contact angles measured on brass not subjected to CT and that treated for 60 min at 120 °C in the vapors of the CIN studied are shown in Table 14.

Compared to the experiments on copper and steel [12], chamber treatment of brass weakly affected the contact angles. The θ angle slightly increased after CT, but this increase was just a little larger than 10° in the case of the ODA + BTA mixture. Larger contact angles were observed after treatment of brass with a pre-created polymodal roughness with the mixed CIN. However, the measured value, θ = 146°, did not allow us to classify this phenomenon as superhydrophobicity.

Brass samples not subjected to thermal treatment corroded 25 days after the start of exposure under conditions similar to those described above (a cardboard box under a shelter at the Moscow Corrosion Station). Chamber treatment of brass for 60 min at 120 °C in the vapors of the mixed inhibitor provided full protection of the metal for more than 25 months. This time is obviously sufficient for the temporary protection of items made of this material. Pictures of samples tested for 25 months under the indicated conditions are shown in Figure 6.

At the time this article was written, the samples still showed no signs of corrosion and their exposure under natural conditions continued.

## 4. Conclusions

A CIN consisting of an ODA + BTA mixture well protects copper and brass from atmospheric corrosion and can be used to temporarily protect metal items.The optimum temperature of CT with the mixed inhibitor is 120 °C for brass and 100 °C for copper. Treatment in its vapors for 1 h at these temperatures creates nanosized adsorption films on the surfaces of these metals. These films stabilize the passive state and provide efficient temporary protection of metal items.The protective effect of the ODA + BTA mixture is superior to that of its components. Assessment of their mutual effects indicates that there is an antagonism of the protective effect on copper and a synergistic enhancement in the case of brass.The results of impedance measurements make it possible to state that the protection of copper and brass by the CIN studied predominantly occurs by using the blocking mechanism.The most significant protective effect of the ODA + BTA formulation was found on brass. This effect is caused by the formation of adsorption films of considerable thickness that have high resistance to ion transfer from the electrolyte to the metal surface and, at the same time, strongly inhibit the Faraday process of metal oxidation.Chamber treatment of the metals studied with the ODA + BTA mixture results in surface hydrophobization that is significant on copper but insignificant on brass. Chamber treatment of copper samples with an artificially created polymodal surface makes the latter superhydrophobic.

## Figures and Tables

**Figure 1 materials-15-01541-f001:**
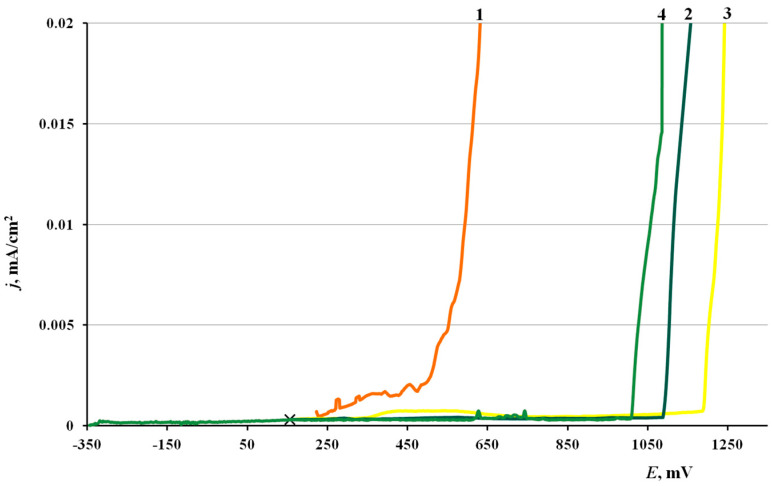
Polarization curves of copper in borate buffer (pH 7.36) +0.001 M NaCl. (1) No treatment with a CIN, (2) BTA, (3) ODA and (4) ODA + BTA.

**Figure 2 materials-15-01541-f002:**
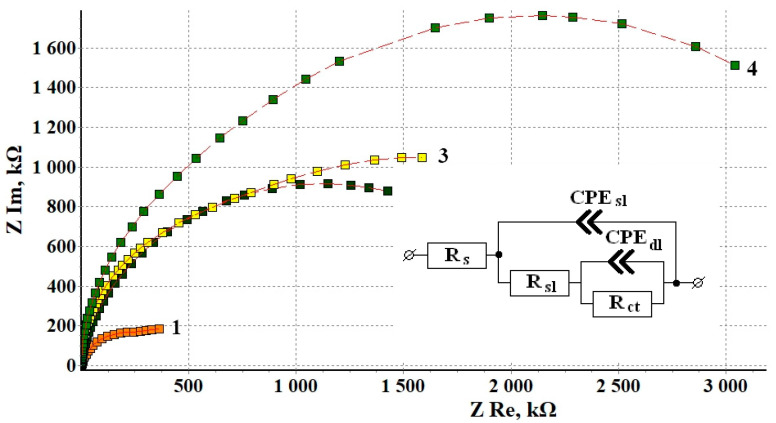
Nyquist plots of copper samples after thermal treatment without a CIN (1) and after CT with the vapors of: BTA (2), ODA (3) and ODA + BTA (4).

**Figure 3 materials-15-01541-f003:**
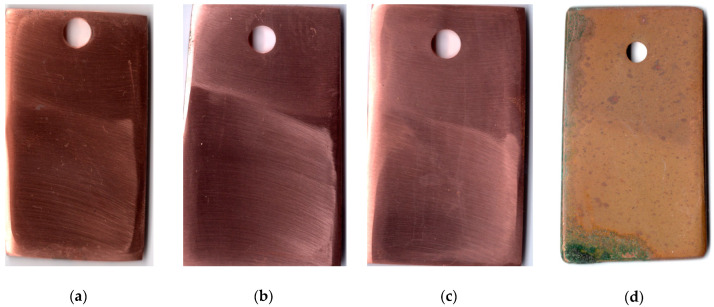
Copper samples after weathering tests for 25 months: chamber treatment with CIN vapors (**a**–**c**); without chamber treatment (**d**).

**Figure 4 materials-15-01541-f004:**
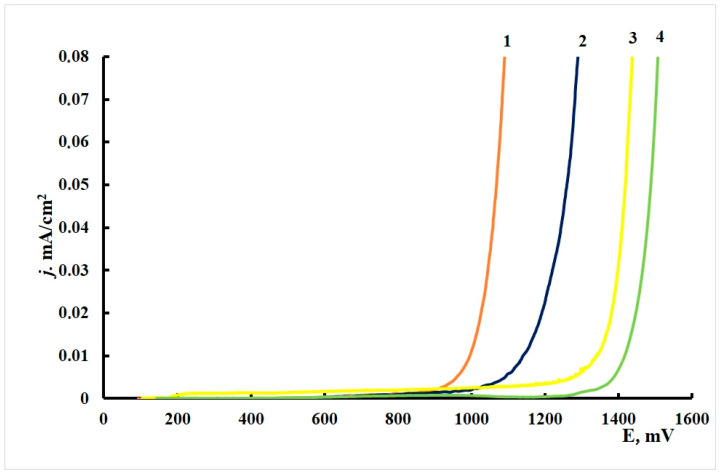
Polarization curves on brass in borate buffer (pH 7.36) + 0.001 M NaCl. (1) Without CIN treatment, (2) BTA, (3) ODA and (4) ODA + BTA.

**Figure 5 materials-15-01541-f005:**
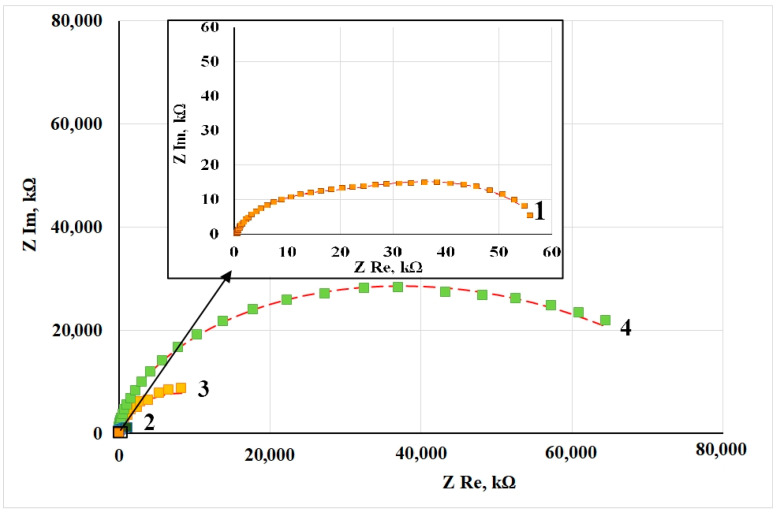
Nyquist plots of brass samples after TT without a CIN (1) and after CT in the vapors of BTA (2), ODA (3) and ODA + BTA (4).

**Figure 6 materials-15-01541-f006:**
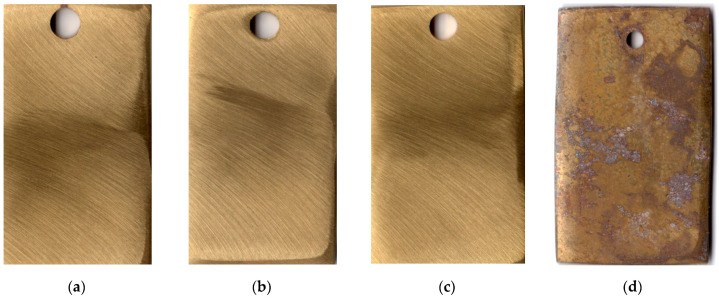
Brass samples after weathering tests for 25 months: chamber treatment with CIN vapors (**a**–**c**); without chamber treatment (**d**).

**Table 1 materials-15-01541-t001:** Protective aftereffect (PAE) of adsorption films formed upon treatment of copper in CIN vapors at different *t* measured in a heat and moisture chamber. τ_CT_ = 60 min.

Chamber Inhibitor	Time until Corrosion Damage Became Visible, h, after Chamber Treatment at the Following Temperatures:			
	80 °C	100 °C	120 °C	140 °C
No CIN	4.0	4.0	- *	- *
BTA	456	960	- *	- *
ODA	288	1080	- *	- *
ODA + BTA	3576	4800	- *	- *

* The samples changed color in the course of thermal and chamber treatment of the metal.

**Table 2 materials-15-01541-t002:** Protective aftereffect of adsorption films formed on copper in CIN vapors at 100 °C and at various τ_CT_ in the salt fog chamber.

Chamber Inhibitor	Time until the Appearance of Corrosion Damage and Cycles after Chamber Treatment for Specified Period		
	20 min	40 min	60 min
No CIN	2	2	2
BTA	15	15	14
ODA	14	17	24
ODA + BTA	15	15	15

**Table 3 materials-15-01541-t003:** Effect of copper chamber treatment on the thickness of surface films. T_CT_ = 100 °C, and τ_CT_ = 60 min.

Chamber Inhibitor	Thickness *d*, nm	
	Oxide Film	Adsorbed CIN Film
No CIN	4.5 ± 0.5	-
BTA	1.5 ± 0.5	2.0 ± 0.5
ODA	1.5 ± 0.5	3.5 ± 0.5
ODA + BTA	1.5 ± 0.5	4.5 ± 0.5

**Table 4 materials-15-01541-t004:** Effect of copper CT on the characteristics of anodic potentiodynamic curves. T_CT_ = 100 °C, and τ_CT_ = 60 min.

Chamber Inhibitor	*E*_cor_, V	*E*_pit_, V	*E*_br_, V
Without a CIN	0.225	0.270	0.510
BTA	0.156	-	1.095
ODA	0.160	-	1.200
ODA + BTA	−0.344	0.600	1.015

**Table 5 materials-15-01541-t005:** Values of equivalent circuit elements based on the results of impedance spectroscopy, as calculated for different variants of copper CT. T_CT_ = 100 °C, and τ_CT_ = 60 min.

Chamber Inhibitor	R_s_ kΩ·cm^2^	R_sl_ kΩ·cm^2^	CPE_sl_ A, Ss^n^/cm^2^	CPE_sl_ n_sl_	R_ct_ kΩ·cm^2^	CPE_dl_ A, Ss^n^/cm^2^	CPE_dl_ n_dl_	Protection Z, %
No CIN	0.4	386.85	9.78 × 10^−7^	0.87	215.57	9.35 × 10^−6^	1	
BTA	0.33	513.66	2.86 × 10^−7^	0.98	1895.94	2.64 × 10^−7^	0.79	89.44
ODA	0.48	1506.41	2.09 × 10^−7^	1	1113.54	9.37 × 10^−7^	1	90.29
ODA + BTA	0.41	2263.29	1.45 × 10^−7^	0.99	1655.61	3.19 × 10^−7^	1	93.51

**Table 6 materials-15-01541-t006:** Inhibition coefficients of chamber inhibitors by the blocking and activation mechanisms after different options of copper CT. T_CT_ = 100 °C, and τ_CT_ = 60 min.

Chamber Inhibitor	γ_sl_	γ_ct_
BTA	1.33	8.80
ODA	3.89	5.17
ODA + BTA	5.85	7.68

**Table 7 materials-15-01541-t007:** Effect of copper CT on the contact angles of wetting with distilled water. T_CT_ = 100 °C, and τ_CT_ = 60 min.

Chamber Inhibitor	Contact Angle, Degree
No CIN	75
BTA	107
ODA	104
ODA + BTA	112
ODA + BTA, polymodal surface	160

**Table 8 materials-15-01541-t008:** Protective aftereffect of adsorption films formed upon brass treatment in CIN vapors at various temperatures. Recurrent moisture condensation conditions. τ_CT_ = 60 min.

Chamber Inhibitor	Time until the Appearance of Corrosion Damage, h, after Chamber Treatment at Temperature:			
	80 °C	100 °C	120 °C	140 °C
No CIN	2.0	2.0	2.0	- *
BTA	2.0	5.0	5.0	- *
ODA	2.0	2.0	2.0	- *
ODA + BTA	24	1416	2232	- *

* The samples changed color in the course of thermal and chamber treatment of the metal.

**Table 9 materials-15-01541-t009:** Protective aftereffect of adsorption films formed on brass in CIN vapors at 120 °C and at various τ_CT_ in the salt fog chamber.

Chamber Inhibitor	Time until Corrosion Damage Appeared, Cycles, after Chamber Treatment for a Period of:			
	15 min	30 min	60 min	75 min
No CIN	1	1	1	- *
BTA	1	1	1	- *
ODA	1	1	1	- *
ODA + BTA	3	5	15	- *

* The samples changed color in the course of thermal treatment or chamber treatment.

**Table 10 materials-15-01541-t010:** Effect of brass chamber treatment on the surface film thickness. T_CT_ = 120 °C, and τ_CT_ = 60 min.

Chamber Inhibitor	Thickness *d*, nm	
	Oxide Film	Adsorbed CIN Film
No CIN	3.25 ± 0.25	-
BTA	3 ± 0.5	19 ± 2
ODA	1 ± 0.5	36 ± 1
ODA + BTA	2 ± 0.5	40 ± 1

**Table 11 materials-15-01541-t011:** Effect of brass CT on the parameters of anodic potentiodynamic curves.

Chamber Inhibitor	*E*_cor_, V	*E*_pit_, V
No CIN	0.105	0.445
BTA	0.095	0.665
ODA	0.110	0.675
ODA + BTA	0.145	1.080

**Table 12 materials-15-01541-t012:** The values of equivalent circuit elements based on the results of impedance spectroscopy calculated for different variants of brass CT. T_CT_ = 120 °C, and τ_CT_ = 60 min.

Chamber Inhibitor	R_s_ KΩ·cm^2^	R_sl_ KΩ·cm^2^	CPE_sl_ A, Ss^n^/cm^2^	CPE_sl_ n_sl_	R_ct_ KΩ·cm^2^	CPE_dl_ A, Ss^n^/cm^2^	CPE_dl_ n_dl_	Protection Z, %
No CIN	0.55	27.5	8.26 × 10^−7^	0.84	31.5	6.22 × 10^−5^	0.76	
BTA	0.47	227.4	3.82 × 10^−7^	1	1701	1.39 × 10^−7^	1	96.94
ODA	0.67	5255	1.47 × 10^−8^	0.98	11833	2.76 × 10^−8^	0.80	99.65
ODA + BTA	0.44	44257	3.61 × 10^−8^	0.97	27284	1.43 × 10^−8^	0.93	99.93

**Table 13 materials-15-01541-t013:** Protective effects of the CINs by the blocking and activation mechanisms after brass chamber treatment in different modes. T_CT_ = 120 °C, and τ_CT_ = 60 min.

Chamber Inhibitor	γ_sl_	γ_ct_
BTA	8.27	54
ODA	191	376
ODA + BTA	1609	866

**Table 14 materials-15-01541-t014:** Effect of brass CT on the contact angles of wetting with distilled water. T_CT_ = 120 °C, and τ_CT_ = 60 min.

Chamber Inhibitor	Contact Angle, Deg.
Without a CIN	85
BTA	88
ODA	92
ODA + BTA	93
ODA + BTA, polymodal roughness	146

## Data Availability

Data sharing is not applicable to this article.

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
