# Peer review of "Mutual Effect of Components of Protective Films Applied on Copper and Brass from Octadecylamine and 1,2,3-Benzotriazole Vapors"

_materials, 2022, doi:10.3390/ma15041541_

Round 1

Reviewer 1 Report

Dear Authors:

(1)The research method and content of this article are highly similar to the author's published paper (Ref. 17). Only the test material steel (Ref. 17) is replaced by copper and brass, which is not innovative;

(2)The test design is too simple. Only the anti-corrosion effects of ODA, BTA, and the mixture of ODA and BTA on copper and brass are tested. The proportion of ODA and BTA used in the test and the reason for this proportion are not described, and the influence of the proportion and dosage of the mixture of ODA and BTA on copper corrosion is not studied; 

(3) This paper is only an experimental report, lacking the mechanism analysis of the synergistic effect and antagonism effect of ODA and BTA, and the research depth is not enough.

Author Response

We are grateful to the Reviewer for valuable comments. By taking them into account, we were able to improve our article considerably. As concerns particular comments:

(1)The research method and content of this article are highly similar to the author's published paper (Ref. 17). Only the test material steel (Ref. 17) is replaced by copper and brass, which is not innovative;

Initially we planned to publish one article to report the chamber treatment of steel, copper, and brass all at once. However, there was too much data to fit in a single article. In view of this, we had to split it into two closely related ones. They should be considered in combination. If they are considered together, we believe that the innovative character of the articles is unquestionable. We emphasized that both articles are related in the coverletter and Introduction of the corrected article version.

(2)The test design is too simple. Only the anti-corrosion effects of ODA, BTA, and the mixture of ODA and BTA on copper and brass are tested. The proportion of ODA and BTA used in the test and the reason for this proportion are not described, and the influence of the proportion and dosage of the mixture of ODA and BTA on copper corrosion is not studied; 

In this article, we studied the effect of the ODA+BTA mixed inhibitor not only on the corrosion properties of copper and brass but also on their electrochemical behavior, thicknesses of surface films, and wetting of metal surfaces with water.

The inhibitor dosage was 0.5 grams per 0.6 L cell. This is indicated in the article concerning steel, and we made a reference to the Experimental section of that article. Apparently, the reference to the experimental technique in another article was not a very good solution for making the article shorter. In the revised version, we expanded the Experimental section considerably.

As concerns the inhibitor dosage, it should be noted that the protective properties of chamber inhibitors do not depend on the amount introduced in the chamber in a very wide range of experimental conditions. The properties and structure of surface films formed upon chamber treatment are determined by the atmosphere inside the chamber, which is saturated with inhibitor vapors. However, the saturated vapor pressure is determined by the inhibitor nature and temperature and not on its amount in a very wide range of conditions. Similar protective properties toward steel, copper, and brass were observed if the dosage was 0.05 or 5 grams per cell.

The mass ratio of ODA and BTA in the mixture was 1:1. We mentioned it in the revised variant of the article. If the mixture components do not interact, as we assumed when planning this study, their ratio should not have affected the atmosphere composition in the chamber and the protective properties of the CIN. However, one cannot rule out the interaction of ODA and BTA in the mixture. The Reviewer is correct that the quantitative composition of the ODA+BTA mixture, at least at temperatures above the melting points of its components, may affect the atmosphere composition and hence the properties of the adsorbed films. The effect of the ratio in ODA+BTA mixtures on the efficiency of chamber protection and properties of adsorption inhibitors will be the subject of our studies in the near future.

(3) This paper is only an experimental report, lacking the mechanism analysis of the synergistic effect and antagonism effect of ODA and BTA, and the research depth is not enough.

As mentioned above, the article should be considered in combination with the publication concerning steel. In such a case we believe that it becomes innovative and deep enough.

As concerns the synergism and antagonism of the components: we are not aware of any publications on vapor-phase inhibitory protection of metals offering convincing explanations of the mechanisms of these processes. The systems are too complex. Moreover, as shown in our article concerning steel, the criteria of the mutual effect of components are not always obvious. We will certainly try to look into the mechanisms of the mutual effect of ODA and BTA in the chamber protection of metals, however, we doubt it can be done very soon.

Reviewer 2 Report

The introduction part is too basic. There are necessary more info with inputs from other authors having approach the subject and underlining the novelty of the present research. The materials and methods section is completely unacceptable. The section should be reformulated and the materials used and methods properly described. The results are interesting but the discussion should be more directed on the behavior of copper and brass under corrosion.

Author Response

We are grateful to the Reviewer for valuable comments. By taking them into account, we were able to improve our article considerably. As concerns particular comments:

The introduction part is too basic. There are necessary more info with inputs from other authors having approach the subject and underlining the novelty of the present research.

We expanded the Introduction considerably and tried to present information showing the novelty of the chamber protection methods and the advantages of chamber inhibitors compared to traditional volatile corrosion inhibitors.

The materials and methods section is completely unacceptable. The section should be reformulated and the materials used and methods properly described.

We adjusted the materials and methods section significantly.

The results are interesting but the discussion should be more directed on the behavior of copper and brass under corrosion

The sections concerning outdoor testing of chamber inhibitors have been expanded.

Reviewer 3 Report

Dear authors,

Reviewer's comments:

  1. The work presents excellent inhibition efficiency of two vapor phase inhibitors (very less work has been done successfully), a mixture of octadecylamine (ODA) and benzotriazole (BTA) that efficiently protects copper and brass from atmospheric.
  2. The ODA+BTA inhibitor is superior to its components in terms of protective aftereffect. Analysis of the mutual effect of BTA and ODA indicates that they show an antagonism of protective action on copper but a synergistic enhancement in the case of brass. (Study of synergism adds extra quality to this paper, and will be very helpful for the readers working in the area of VCI’s).
  3. Accelerated corrosion tests provide great support to author’s claim, especially when the work has been done for 3576 hours at 100 o So, a tremendous effort to provide some useful data in corrosion testing and that makes it worth publishing.
  4. Ellipsometry and Voltammetric studies are another good inclusion in the paper and are well presented along with the data in Table.
  5. Impedance studies (EIS) shows perfect use of corresponding circuit and hence the data in table matches perfectly.
  6. Likewise contact angle tests also prove the hydrophobhicity in presence of VCI’s.

Overall a good paper. Just improve the quality of figures and make the Nyquist graphs in Impedance studies orthogonal (x and y with same axes). So, the work is really well written and provides a great deal of information to readers. I highly recommend for its publication.

Author Response

We are grateful to the Reviewer for the high opinion about our work. As concerns the comment:

Just improve the quality of figures and make the Nyquist graphs in Impedance studies orthogonal (x and y with same axes)

We tried to improve the quality of the figures.

Round 2

Reviewer 1 Report

In Table 2 and table 9, the time unit is' h ', while in line 289 or 478, the time unit is' cycles', which are inconsistent. Please correct.
In Table 2 and table 9, the CT time is 60min, while in line 297, 305 or 497, the CT time is 1 hour, and the CT time unit is preferably the same in both places.
It is not necessary to enlarge part of Figure 3.

Author Response

Thank you very much for your comments!

In Table 2 and table 9, the time unit is' h ', while in line 289 or 478, the time unit is' cycles', which are inconsistent. Please correct.

“Cycles” are now indicated as the measurement unit for the protective effect of CIN under salt fog conditions.

In Table 2 and table 9, the CT time is 60min, while in line 297, 305 or 497, the CT time is 1 hour, and the CT time unit is preferably the same in both places.

The chamber treatment duration has been converted to minutes throughout the article.

It is not necessary to enlarge part of Figure 3.

The PDF and DOCX versions of the article available on the site (https://susy.mdpi.com/user/manuscripts/resubmit/a331a97cf53fffe03445d3e08810ac4c) differ significantly, apparently because tracked changes were not accepted before creating the PDF. As a result, both new and old versions of changed Figures are present in the PDF. In the DOCX version, part of Figure 3 is not enlarged.

Reviewer 2 Report

The changes were performed.

Author Response

Thank you very much for your comments!